# Stop Codon Context-Specific Induction of Translational Readthrough

**DOI:** 10.3390/biom11071006

**Published:** 2021-07-09

**Authors:** Mirco Schilff, Yelena Sargsyan, Julia Hofhuis, Sven Thoms

**Affiliations:** 1Department of Child and Adolescent Health, University Medical Center, 37075 Göttingen, Germany; mirco.schilff@stud.uni-goettingen.de (M.S.); yelena.sargsyan@stud.uni-goettingen.de (Y.S.); julia.hofhuis@uni-bielefeld.de (J.H.); 2Department of Biochemistry and Molecular Medicine, Medical School, Bielefeld University, 33615 Bielefeld, Germany

**Keywords:** translational readthrough, rare disease, peroxisome, peroxisome biogenesis disorder, PEX5, personalized medicine, readthrough therapy, aminoglycoside

## Abstract

Premature termination codon (PTC) mutations account for approximately 10% of pathogenic variants in monogenic diseases. Stimulation of translational readthrough, also known as stop codon suppression, using translational readthrough-inducing drugs (TRIDs) may serve as a possible therapeutic strategy for the treatment of genetic PTC diseases. One important parameter governing readthrough is the stop codon context (SCC)—the stop codon itself and the nucleotides in the vicinity of the stop codon on the mRNA. However, the quantitative influence of the SCC on treatment outcome and on appropriate drug concentrations are largely unknown. Here, we analyze the readthrough-stimulatory effect of various readthrough-inducing drugs on the SCCs of five common premature termination codon mutations of *PEX5* in a sensitive dual reporter system. Mutations in *PEX5*, encoding the peroxisomal targeting signal 1 receptor, can cause peroxisomal biogenesis disorders of the Zellweger spectrum. We show that the stop context has a strong influence on the levels of readthrough stimulation and impacts the choice of the most effective drug and its concentration. These results highlight potential advantages and the personalized medicine nature of an SCC-based strategy in the therapy of rare diseases.

## 1. Introduction

Rare diseases pose a challenge to society as they are often diagnosed late and therapeutic options are scarce. In Europe, a disease is defined as rare if it affects less than 1:2000. In the United States, a rare disease is defined as a condition that affects fewer than 200,000 people, which is equivalent to 1:1640, based on a population of 328 million. The estimated number of rare diseases ranges between 7000 and 10,000 [1]. Thus, contrary to what the name implies, rare diseases in total are not rare, affecting approximately 350 million patients worldwide, far outnumbering e.g., the number of cancer patients [2].

Generally, therapeutic approaches to rare disease treatment include antibody therapies, protein replacement, oligonucleotides, gene therapy, small molecule therapies, and drug repurposing [2]. The majority of rare diseases are inheritable and monogenetic. Nonsense (stop codon) mutations, also referred to as premature termination codon (PTC) mutations, are responsible for about 10% of all cases of monogenetic diseases [3]. One of the currently discussed therapeutic options for the treatment of diseases caused by nonsense mutations is the induction of translational readthrough (TR). In TR, the stop codon is read as an amino acid-encoding (sense) codon, so that the translation continues and often the functional protein gene product is regained. Readthrough as a therapeutic option for genetic diseases caused by nonsense mutations is examined for several diseases, including cystic fibrosis and Duchenne muscular dystrophy [4,5].

Readthrough-inducing drugs act at the ribosome and shift the equilibrium between termination and (mis)reading of the stop as a sense codon towards the latter. The aminoglycosides geneticin (G418) and gentamicin are two potent and often used drugs inducing miscoding and readthrough [6,7,8]. Although it is an advantage that aminoglycosides have been in clinical use for years (drug repurposing), their application is limited due to their oto- and nephrotoxicity [9,10,11]. In order to reduce the toxic side effects and to increase their readthrough-inducing potential, a series of aminoglycoside-derivatives are being developed and examined [12,13,14,15]. Ataluren (formerly PTC124) is a non-aminoglycoside small molecule drug approved for treatment of certain Duchenne muscular dystrophy cases in several countries [16].

Independently of pharmacological stimulation, readthrough occurs naturally, as a consequence of inevitable translational inaccuracy. It has been found that certain nucleotides in the vicinity of the stop codon support endogenous readthrough. Such elevation of basal readthrough has been exploited in evolution and led to the notion of ‘programmed TR’, or, if the readthrough-derived protein extension bears a function that is not present in the parent protein, of ‘functional TR’ [17,18]. The level of programmed/functional readthrough is highly dependent on the stop codon (with UGA being the leakiest stop codon) and some of the nucleotides upstream and downstream of the stop codon. The combination of stop codon together with the ten or so nucleotides in its vicinity is termed stop codon context (SCC). The SCC, particularly the nucleotide directly following the stop codon, and the stop codon itself highly influence the degree of readthrough [19,20,21]. For example, the UGA stop codon followed by a C present an especially ‘leaky’ stop context [20,22,23].

Not only endogenous readthrough, but also stimulated readthrough is dependent on the SCC. Therefore consideration of the stop context of a given PTC could allow the development of a more suitable therapeutic approach for a patient. This approach of addressing the disease and its treatment in a patient genotype-specific manner could be at the basis of personalized medicine [24,25] of PTC diseases.

Nonsense mutations are a potential cause of peroxisomal biogenesis disorders (PBD), rare inheritable diseases also known as Zellweger spectrum disorders that are based on dysfunctional peroxisome assembly. Peroxisomes are cellular organelles essential for human health as they play major roles in lipid and hydrogen peroxide metabolism. The most severe form of these diseases is termed Zellweger Syndrome (ZS). Patients with ZS present with severe and often fatal cerebral, renal, and hepatic symptoms. Defects in any of the 16 human PEX proteins can be causative for ZS. *PEX1* is affected in more than 50% of the cases, and roughly 2% of all ZS are due to pathogenic variants of *PEX5* gene [26]. The PEX5 protein is the peroxisomal targeting signal 1 (PTS1) receptor that recognizes the C-terminal PTS1 sequence of cargo proteins and mediates the PTS1-dependent import into the peroxisomal matrix. Pathogenic variants of PEX5 cause a disturbed import of peroxisomal matrix proteins and lead to ZS [27,28]. Among the *PEX* pathogenic nonsense mutations, so far only *PEX2* and *PEX12* variants have been evaluated with regard to TR induction potential as a therapeutic option [29].

In this work, we analyze the SCCs of four of the most common *PEX5*-nonsense mutations [26] and of one mutation recently identified by us [30]. We demonstrate that basal readthrough and readthrough induction depend on the SCC. All tested drugs have a context-dependent effect on TR induction and their effective dosage is different for each SCC. The stop codon context also affects the choice of a suitable readthrough-inducing drug. Hence, readthrough therapeutic approaches should consider the combination of SCC and drug. A context- and drug-focused approach may serve as a basis for a personalized medicine strategy for the treatment of diseases caused by nonsense mutations.

## 2. Materials and Methods

### 2.1. DNA Constructs and Cloning 

Dual reporter constructs for flow cytometry were cloned based on the dual reporter vector described before [31]. The dual reporter vector (PST 1596) contains red fluorescent protein (RFP) in frame with green fluorescent protein (GFP) in the pcDNA3.1/Zeo (+) backbone. This was also used as positive (100% readthrough) control for TR calculation. For detection of background FITC values, an RFP-containing construct was used (PST 1880). Five dual reporter plasmids containing ZS-causing *PEX5*-nonsense mutations were constructed. Oligonucleotides containing the PTC stop contexts (ten nucleotides upstream and downstream of the PTC) and BspEI and BstEII restrictions sites were synthesized. After oligonucleotide annealing, the insert was ligated into the dual reporter backbone [31]. Resulting constructs were checked by DNA sequencing.

The following oligonucleotides were used for cloning of the SCC dual reporters: PST 2185 (c.397 C>T; p.Q133X): 5′-GTCAC CGGGA TGTAA CTTAG GATTA TAATG T-3′ (OST 2231), and 5′-CCGGA CATTA TAATC CTAAG TTACA TCCCG GTCG-3′ (OST 2232); PST 2186 (c.826 C>T; p.R276X): 5′-GTCAC CGGGA GTTTG AATGA GCCAA GTCAG T-3′ (OST 2233), and 5′-CCGGA CATTA TAATC CTAAG TTACA TCCCG GTCG-3′ (OST 2234); PST 2187 (c.1090 C>T; p.Q364X): 5′-GTCAC CGAGC TGTGC AGTAG GATCC TAAGC T-3′ (OST 2235), and 5′-CCGGA GCTTA GGATC CTACT GCACA GCTCG-3′ (OST 2236); PST 2188 (c.1258 C>T; p.R420X): 5′-GTCAC CGGTC CCTGC AGTGA CAGGC CTGTG T-3′ (OST 2237), and 5′-CCGGA CACAG GCCTG TCACT GCAGG GACCG-3′ (OST 2238); PST 2189 (c.1279 C>T; p.R427X): 5′-GTCAC CGTGA AACCC TATGA GACTG GCTGC T-3′ (OST 2239), and 5′-CCGGA GCAGC CAGTC TCATA GGGTT TCACG-3′ (OST 2240). 

### 2.2. Cell Culture

HeLa cells were cultured at 37 °C and 5% CO_2_ in low glucose Dulbecco’s minimal essential medium (DMEM) supplemented with 10% heat-inactivated fetal calf serum (FCS), 1% glutamine, 100 units/mL penicillin and 100 µg/mL streptomycin. 1 × 10^5^ cells were seeded per well in a 12-well plate and transfected using Effectene transfection reagent (Qiagen) according to the manufacturer’s instructions. Transfection reagent was removed six hours after transfection. The cells were treated with the respective TRIDs and incubated for another 18 h. Amikacin, erythromycin, josamycin, paromomycin, tobramycin and tylosin were obtained from Sigma-Aldrich, gentamicin and G418 from Carl Roth. Stock solutions were made in water, except for erythromycin (70% ethanol). 

### 2.3. Flow Cytometry-Based Dual Reporter Assay and Translational Readthrough Calculation

The flow cytometric dual reporter assay was performed as described before [31]. Briefly, cells from one well were washed with 100 µL phosphate-buffered saline (PBS), trypsinized, and incubated for five minutes. The cells were then resuspended in 1 mL DMEM, pelleted (1000 rpm, 5 min), resuspended in 400 µL PBS with 5% FCS and then filtered through a 70 µm cell strainer into the flow cytometry tubes. Flow cytometry (Becton Dickinson LSR II) was carried out with 488 nm and 561 nm lasers using FACS Diva software (Becton Dickinson). For readthrough calculation, gates for forward scatter were set between 70,000 and 180,000 and side scatter between 50,000 and 150,000. All cells with RFP values (PE-A) above 220 or GFP values (FITC-A) above 100 were included in the analysis. For each replicate, a 100% TR control vector (PST 1596) only containing RFP and GFP was measured. TR was then calculated by subtracting background FITC values (measured with the RFP-containing construct, PST 1880) from gated FITC values. The ratio of FITC and PE values was calculated, and the ratio of the sample was divided by the ratio of the 100% readthrough control vector to normalize TR. Readthrough prediction was done by using a bioinformatics prediction tool published earlier [20]. 

### 2.4. Statistics

Statistical analysis was performed with GraphPad Prism using ordinary one-way ANOVA. *p*-values less than 0.05 were considered as statistically significant. Figures were created using GraphPad Prism. If not stated otherwise, data was presented showing the median, minimum and maximum values and individual data points.

## 3. Results

### 3.1. The Stop Codon Context Affects Basal Translational Readthrough and Induced Translational Readthrough in PEX5 Nonsense Mutations

To measure translational readthrough, we used a flow cytometry dual reporter assay with reporter-transfected HeLa cells [31]. The vector contains an RFP sequence upstream and a GFP sequence downstream of the stop codon context, so that TR can be calculated as the ratio of GFP and RFP fluorescence after flow cytometry measurement (Figure 1A). As the wider SCC context does not have significant influence on TR [17,21], we focused on analyzing the 10 nucleotides up- and downstream the stop codon. To analyze the readthrough of *PEX5*-nonsense mutations, five SCCs of *PEX5*-nonsense mutations were cloned into the dual reporter vector (Figure 1A,B). The stop mutations c.826C>T (GAA TGA GCCA), c.1090C>T (CAG TAG GATC), c.1258C>T (CAG TGA CAGG), c.1279C>T (CTA TGA GACT), belong to common pathogenic *PEX5* alleles [26], and the first c.397 C>T (ACT TAG GATT) patient was recently described by us [30]. Three of these PTC alleles have the readthrough-prone UGA stop codon, which in one case is followed by a C (c.1258), the other two alleles carry the UAG stop codon (Figure 1B).

Before starting TR measurements, we computationally calculated the TR propensity values for the five SCCs using the regression modeling prediction tool that we developed earlier [20]. Predicted TR was with up to 0.3 the highest for c.1258C>T (UGA C) and the lowest for the two UAG codons (c.379C>T and c.1090C>T) (Figure 1C).

Readthrough and readthrough induction of the *PEX5* SCCs were then measured in HeLa cells by flow cytometry. For each measurement 10,000 events were recorded (Figure 1D–F). Basal TR was lower than 0.2% in all cases (Figure 1G). The c.1258C>T mutation with about 0.15% showed a significantly higher basal readthrough than the other stop codon contexts. In all other *PEX5*-variants basal readthrough was about 0.1%. Only in c.1258C>T the UGA stop codon is followed by a C, a combination known to stimulate high basal readthrough [19,20,32,33]. 

Next, we analyzed the readthrough after pharmacological induction. Treatment with 100 µg/mL G418 induced significant levels of TR at all SCCs (Figure 1H). Yet, the levels of induced readthrough were not identical, and displayed similar trends like the basal TR. The c.1258C>T variant showed with 6% the highest TR induction. This is in accordance with the previous knowledge that C in position 4 in combination with UGA stop codon is the leakiest context prone to readthrough induction [19,34]. Under the same conditions the c.1090C>T variant led to the lowest induction level of about 2% (Figure 1H). These results confirm that the high TR context UGA C has high influence on TR and also on TR induction, but other SCCs can have similarly high induction potential. Further, our results show that not the stop codon alone determines the level of TR induction as not all UGA contexts show higher TR induction than UAG contexts. 

We then calculated the readthrough induction factors, defined as the ratio of induced and basal TR. The c.1258C>T and c.1279C>T stop codon contexts showed highest induction factors with 30.4 ± 6.5 and 32.5 ± 4.6 (mean ± SEM), respectively (Figure 1I).

### 3.2. Dose-Dependent Induction of Translational Readthrough by G418

In the following experiments, we examined readthrough induction of *PEX5* nonsense alleles by increasing concentrations of G418 up to 170 µg/mL. Readthrough of all mutations increased in a dose-dependent manner (Figure 2A–E). G418 induced TR at surprisingly low concentrations. In the c.397C>T construct, 30 µg/mL G418 stimulated readthrough to about 2% (Figure 2D) whereas in the c.1258C>T variant concentrations as low as 5 µg/mL led to induced readthrough up to 4% (Figure 2B). This further underlines the importance of the stop context for the readthrough induction.

Generally, concentrations above 30 µg/mL did not further increase the effect in a proportional manner; an additional increase of concentration would not be beneficial (Figure 2A,B,D,E). Only for the c.1279C>T construct, we found that higher G418 concentrations were accompanied by an additional increase of readthrough (Figure 2C).

In order to relate the readthrough and drug concentration quantitatively we calculated the G418 concentrations required to achieve 1% readthrough and the maximum readthrough values. The required concentrations of G418 for 1% TR varied between 0.76 µg/mL for c.1258C>T and 87.9 µg/mL for c.1090C>T, that is by a factor of more than ten (Figure 2F). The c.1090C>T needs significantly more G418 to reach 1% TR as compared to c.397C>T, both containing UAG followed by GAU. This underlines the importance of the SCC for TR induction.

The maximum readthrough values were assessed by calculating TR for 200 µg/mL G418. Figure 2G shows that these values cover a broad range from 1.1% for c.1090C>T up to 5.4% for c.1258C>T. Collectively, our results indicate a surprisingly large dependence of readthrough on the stop context. This is illustrated by induction with 100 µg/mL G418 (Figure 1F), by 1% TR inducing drug concentrations, and by the maximum readthrough (Figure 2F,G). Hence, in clinical setting, the decisions if a treatment could be beneficial and what would be the optimal drug concentration are expected to be highly dependent on the stop context.

### 3.3. The Effect of Readthrough Induction Depends on the Stop Codon Context

In the next step, we aimed to study the influence of different readthrough drugs on the TR at *PEX5*-nonsense mutations. First, we used the aminoglycosides gentamicin and paromomycin. In all examined constructs both drugs induced readthrough (Figure 3) in a dose-dependent manner when tested with increasing concentrations up to 500 µg/mL (Figure 3A–E). Generally, gentamicin and paromomycin were less potent than G418. Readthrough induction for the SCCs c.397C>T and c.1258C>T by paromomycin was at a comparable or, for c.397C>T even slightly higher level than gentamicin using concentrations of 100 µg/mL and 500 µg/mL, respectively (Figure 3D). All other mutations showed higher levels of TR by induction with gentamicin (Figure 3A–C,E). For c.826C>T the highest readthrough was seen with 500 µg/mL gentamicin treatment (Figure 3A). The lowest TR levels were found for c.1090C>T, with the most prominent effect with 500 µg/mL gentamicin (Figure 3E). This variant had the lowest readthrough levels also after G418 treatment. Upon treatment with gentamicin and paromomycin, the SCC c.1258C>T showed the highest TRs compared to other mutations (Figure 3B). Gentamicin induced readthrough between 0.2% (Figure 3E) and 1% (Figure 3B) with 100 µg/mL. At the same concentration, paromomycin induced TR from 0.1% (Figure 3E) to 0.5% (Figure 3B). When increasing the concentration up to 500 µg/mL, gentamicin induced between 0.5% and 1.5% readthrough (Figure 3E,B) while the induction by paromomycin ranged between 0.2% and 1% (Figure 3A,D). 

In all tested conditions, mutations c.397C>T and c.1258C>T showed the highest responses, therefore we further tested them with a set of substances that were reported to act as readthrough-inducing drugs: the two additional aminoglycosides amikacin and tobramycin and the macrolides erythromycin, josamycin and tylosin [4,7,35].

The c.397C>T construct was treated with amikacin and tobramycin concentrations of 100 µg/mL and 500 µg/mL. Tobramycin failed to show readthrough at 100 µg/mL but we observed a minute increase above basal TR at 500 µg/mL (Figure 3F). Amikacin at 500 µg/mL induced readthrough up to 0.25% for c.397C>T (Figure 3F). Therefore, we decided to test all five SCCs with 500 µg/mL amikacin and found that c.1258C>T was inducible up to 0.3%, which is not significantly higher compared to c.397C>T (a UAG context) but higher compared to all other tested SCCs (Figure 3G). Remarkably, induced TR of c.397C>T is even higher than induced TR of c.826C>T, a UGA context (Figure 3G). The macrolides erythromycin, tylosin, and josamycin did not induce TR at the two SCCs tested, c.1258C>T and c.397C>T (Figure 4A,B). Readthrough induction by tylosin was not measurable at 500 µg/mL due to its cytotoxicity. 

Taken together, our results show that pharmacological readthrough induction depends on the stop codon context in a double sense. On one hand, readthrough induction and the maximum TR are dependent on the SCC. On the other hand, and more surprisingly, we found that the stop codon context may influence the choice of a translational readthrough-inducing drug. These results reflect the personalized medicine nature of any translational readthrough-based approach.

## 4. Discussion

Genetic diseases caused by PTC are found in all clinical disciplines. The pharmacological treatment of such diseases could involve readthrough therapy. In experimental systems, the degree of TR stimulation is often between 1% and 10% of the wild-type expression, but even levels lower than 1% might be therapeutically beneficial [36]. However, the molecular determinants of readthrough are still unclear [19,20,37]. Our previous study in which we formalized and modelled the influence of the SCC by machine learning using a broad range of sequence variants indicates that responses to readthrough and readthrough induction are highly variable and depend on the stop codon and its nearest nucleotide context [20]. This is in line with several other studies [12,19,20,35]. The current study focusses on SCC-dependency of readthrough induction of the *PEX5* PTCs.

### 4.1. Stop Codon Context-Dependency of Readthrough Induction

In our experiments, basal and stimulated readthrough varied between the *PEX5* pathogenic variants, highlighting the impact of the stop codon context on TR. Traditionally, the SCC showing the highest readthrough (basal and induced) contains the UGA stop codon followed by C [20,22,23] which is in line with our TR prediction. For the two UGA G constructs, predicted TR was higher than TR of the two UAG constructs. The experimental results, however, show that UGA G and UAG readthrough were not distinct, indicating that despite the known influences of the stop codon and position 4 on TR, additional positions or combinations of positions influence TR. TR induction by G418 was not significantly different between most of the SCCs highlighting that other SCCs might have a comparable potential of TR induction compared to the high TR context UGA C and that even UAG contexts can be similarly effective induced as UGA contexts. 

We found that comparably low concentrations of G418 could induce high levels of readthrough and that readthrough stimulation could be saturated [17,37,38]. The maximum readthrough induction level, and, at a given drug concentration, the readthrough level and the induction factor may serve as important parameters when deciding for a TR-based therapeutic approach. Dual reporter assays may help to decide if such a therapeutic strategy could be taken into consideration. The dual reporter assay allows optimization of a drug’s concentration for a specific stop context. In case of readthrough therapy, it would be additionally useful to know the level of readthrough that is needed to restore the protein function. This differs between diseases and even values below 1% of TR might be sufficient for a therapeutic effect [4,39].

Our results additionally show that the stop codon context may also affect the choice of the readthrough-inducing drug. When comparing gentamicin and paromomycin, we found that only for the c.397C>T and c.1258C>T constructs paromomycin was as effective as gentamicin (Figure 3). Also, amikacin was able to induce similar amounts of TR for c.397C>T compared to c.1258C>T and even higher levels of TR compared to c.826C>T outlining the importance of the SCC for the choice of TRID. These results indicate that TRIDs have distinct effects on TR induction regarding different stop codon contexts. In other words, the SCC not only dictates the optimal drug concentration, but also influences the optimal drug itself.

Taken together, our study points out the influence of the SCC on basal and induced TR. It conveys as well that the SCC has an impact on the specific TR that could be achieved by a suitable drug in an effective concentration. 

### 4.2. Readthrough Stimulation as a Therapeutic Option for PBD

For peroxisomal biogenesis disorders, premature termination codon mutations in *PEX2* and *PEX12* have previously been analyzed showing improved metabolic activity after G418 treatment [29]. There are many challenges, however, that must be addressed before the added specificity of an SCC-focused approach can be translated to therapy. Further experiments would be necessary to analyze the readthrough of *PEX5*-nonsense mutations, the production of the full-length protein and its function as a PTS1 receptor. These experiments, when done in cellular models, require genomically altered *PEX5*, as overexpression of PEX5 acts in a dominant negative way that would overwrite the readthrough effect. There are not only the limiting side effects of existing readthrough-inducing drugs, a therapy may also fail because the stop-codon-encoded amino acid does not restore functionality, or the nonsense-mediated mRNA decay (NMD) degrades the mRNA leaving no substrate for the induction of TR [40,41,42,43]. The latter can be overcome by combination of TRID with proteasome modulator CC-90009, a combination that enhances the stability of gene transcripts of several PTC and enhances TR efficiency [44]. Further increase of TR can be achieved by the stimulation of ribosome biogenesis with Y-320 in combination with TRIDs [45]. The eventual clinical outcome of TR therapy may also depend on the main organ system affected, as readthrough may vary in different tissue types [46,47]. The current study suggests an uncomplicated in vitro screening for the mutations that could be considered for more in vivo and more clinical analyses of TR efficiency.

### 4.3. Readthrough Therapy as Personalized Medicine 

The stop codon context is an important determinant and possibly even a predictor of readthrough inducibility [21]. Therefore, knowledge about the exact genetic situation is essential for a suitable TR-based therapeutic approach. There is thus an inherent, previously underappreciated heterogeneity in PTC-diseases in this regard. The genetic background and efficiency of readthrough induction along with the choice of the appropriate drug may serve as basis of personalized medicine strategy in the treatment of diseases caused by premature stop codon mutations. The strategy of personalized medicine that, among other factors, considers the genetic background was a breakthrough in the treatment of neoplasms and is currently successfully used in the treatment of leukemias and lymphomas [48,49,50].

A personalized therapeutic strategy including SCC-adapted optimization of a drug’s concentration such as G418 could promote the therapeutic use of aminoglycosides and help to reduce their side effects by finding the minimal effective dose for each stop context and thus follow a longer systemic treatment. Further, the toxic side effects may be reduced by combination of other strategies such as different pharmaceutical formulations and effective dose intervals [51] or co-application of drugs which could reduce the toxic side effects [52,53]. 

Designer aminoglycosides offer another option to reduce aminoglycoside toxicity. They contain the molecular structures which are thought to be responsible for readthrough induction [54] but lack the structures causing toxicity. Some of them already showed TR induction [8,55,56,57]. ELX-02 (formerly NB124), a non-antibiotic aminoglycoside analog, showed well-tolerance in clinical trials and restoration of functional protein in cystic fibrosis models, presenting a perspective TRID and gentamycin alternative for treatment of PTC-caused diseases [15,58]. Another option to bypass the toxicity of aminoglycosides is the use of non-aminoglycoside TRIDs. Therefore drugs such as ataluren were developed [14]. Though showing at least some readthrough induction and benefits in several genetic disease models besides Duchenne muscular dystrophy (for which it is currently accepted treatment option) [47,59,60], ataluren fails to induce readthrough in others [47,61,62,63].

In summary, our work emphasizes the special importance of the stop context for readthrough and its stimulation. We show that the SCC influences basal readthrough levels and the induction of readthrough by treatment with a specific drug. Consequently, it is crucial to know the effect of a specific TRID in an effective dosage on a specific stop codon context which would improve the therapeutic benefit. This work shows that a more differentiated analysis by simple means can help stratify patients according to treatment chances. The knowledge about the nonsense mutation itself with its context is a prerequisite (but not a sufficient condition) for effective therapy so that patients with any genetic disease caused by a PTC profit from this personalized medicine approach.

## Figures and Tables

**Figure 1 biomolecules-11-01006-f001:**
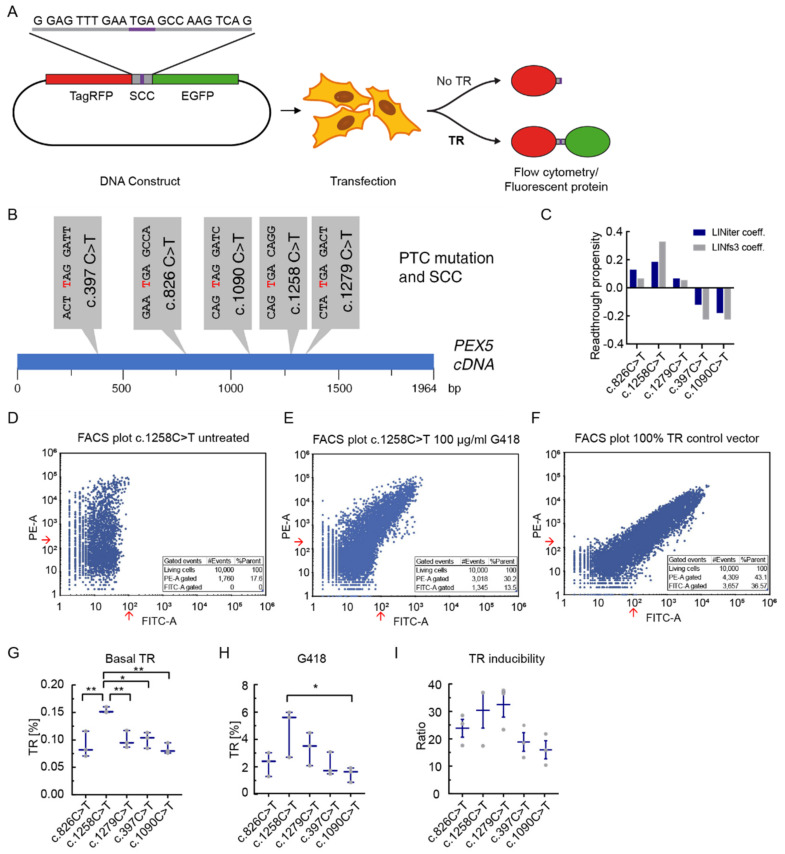
The SCC affects basal and induced translational readthrough. (**A**) Dual reporter TR assay. The vector contains an SCC between an RFP- and a GFP-sequence. TR is assessed by flow cytometry after transfection. (**B**) *PEX5* SCCs analyzed in this study. The mutated nucleotide resulting in PTC is shown in red. (**C**) TR propensity values for the five SCCs was predicted according to LINiter and LINfs3 model from [20]. (**D**–**F**) Representative FACS plot and table of events per gate. Red arrows: Gates for TR were 220 for RFP (‘PE-A’) and 100 for GFP (‘FITC-A’). 10,000 events were recorded per measurement. (**G**) Flow cytometry-based dual reporter assay of *PEX5*-SCCs shows that basal TR differs between different SSCs. (**H**) G418 is a potent TRID and induces significant levels of TR in all SCCs at a concentration of 100 µg/mL (Ordinary one-way ANOVA, Tukey’s multiple comparisons test induced vs. basal TR: c.826C>T: *p* = 0.026, c.1258C>T: *p* = 0.00024, c.1279C>T: *p* = 0.021, c.397C>T: *p* = 0.00022, c.1090C>T: *p* = 0.025). TR induction depends on the SCC and differs between the SCCs. (**I**) TR induction factor (Ratio = stimulated/basal TR) shows SCC dependency. Mean ± SEM. N = 3 (**E**–**G**), * *p* < 0.05, ** *p* < 0.01 (Ordinary one-way ANOVA, Tukey’s multiple comparisons test).

**Figure 2 biomolecules-11-01006-f002:**
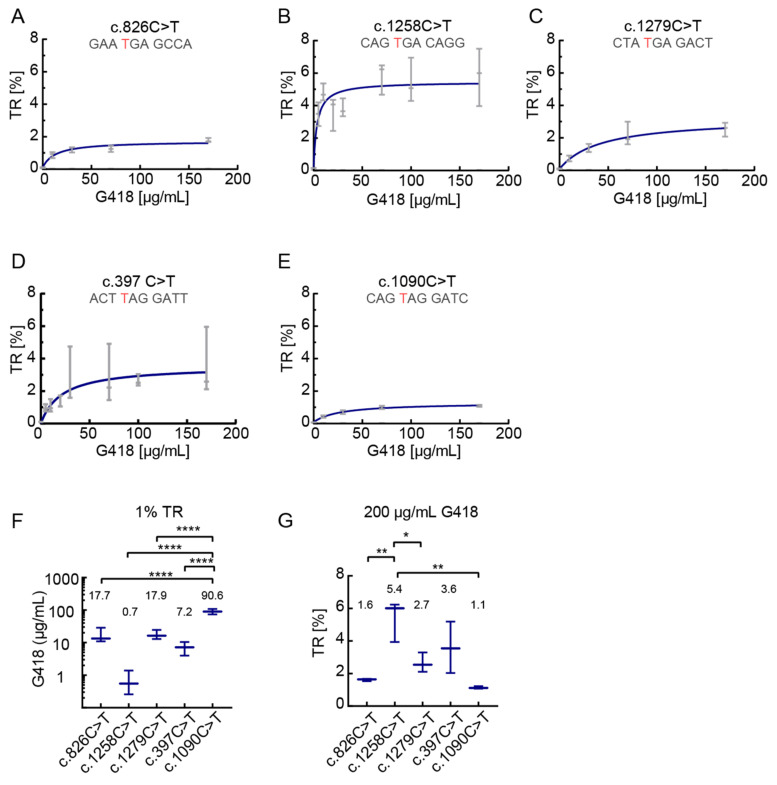
Dose-response analysis of SCC-dependent translational readthrough. G418 was used for TR induction at concentrations up to 170 µg/mL. All SCCs showed a dose-dependent increase of TR. Maximal TR induction differs between SCCs indicating SCC dependency. (**A**) The highest TR observed in c.826C>T is less than 2%. N = 3. (**B**) G418 induces high levels of TR at low drug concentrations. Maximum TR is approximately 5%, higher than in the other tested SCCs. N = 3. (**C**) Only in c.1279C>T TR does not reach a plateau at 170 mg/mL G418. (**D**) Even low concentrations of G418 induce TR. N = 3 (for 30, 70 and 170 µg/mL N = 5). (**E**) Induction by G418 at concentrations > 30 µg/mL further increases TR levels. N = 3. (**F**) 1% TR can be reached with different amounts of G418. Calculated from (**A**–**E**). (**G**) 200 µg/mL G418 induce different levels of TR. Calculated from (**A**–**E**). (**F**,**G**) Data shown as mean and 95% confidence interval. * *p* < 0.05, ** *p* < 0.01, **** *p* < 0.0001 (Ordinary one-way ANOVA, Tukey’s multiple comparisons test). Nucleotide variation resulting in PTC is shown in red.

**Figure 3 biomolecules-11-01006-f003:**
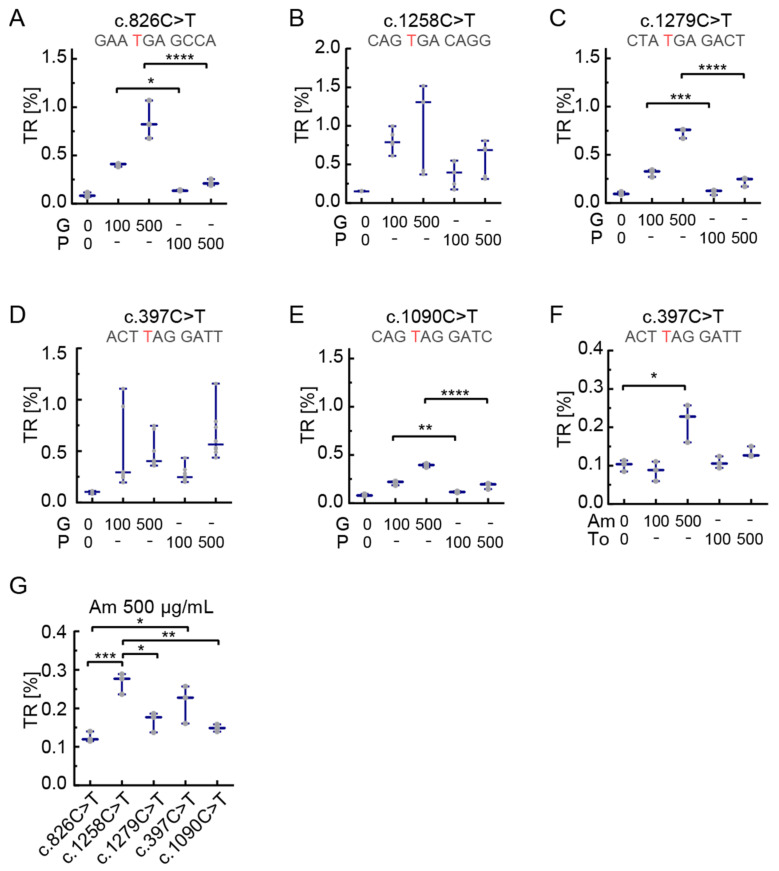
SCC-dependent translational readthrough induction by aminoglycosides. TR induction of gentamicin (G) and paromomycin (P) was analyzed at concentrations of 100 and 500 µg/mL. (**A**–**C**,**E**) Gentamicin leads to higher TR (ns for B) induction than paromomycin in all tested SCCs except c.397C>T. (**D**) In contrast to the other SCCs, paromomycin induces similar TR at c.397C>T as gentamicin. (**F**) Amikacin (Am) at 500 µg/mL induced significant TR (about 0.25%), while tobramycin (To) did not induce significant levels of TR. (**G**) At 500 µg/mL, Amikacin induced TR best in c.1258C>T and the c.397C>T SCC. These levels were significantly higher than for the other three SCC. N = 5 (**A**,**D**); N = 3 (**B**,**C**,**E**–**G**); * *p* < 0.05, ** *p* < 0.01, *** *p* < 0.001, **** *p* < 0.0001 (Ordinary one-way ANOVA, Tukey’s multiple comparisons test). Concentrations in µg/mL. The mutated nucleotide resulting in PTC is shown in red.

**Figure 4 biomolecules-11-01006-f004:**
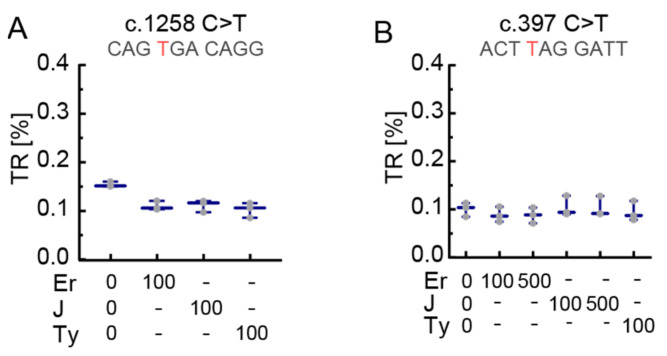
Absence of readthrough induction in *PEX5* SCC reporters by macrolides. (**A**) In the c.1258C>T SCC, which showed highest levels of TR induction by G418 and gentamicin, neither erythromycin (Er), nor josamycin (J), nor tylosin (Ty) induced significant levels of TR. (**B**) Erythromycin, josamycin or tylosin at concentrations of 100 and 500 µg/mL did not lead to significant levels of TR in c.397C>T SCC. N = 3. Concentrations in µg/mL. Nucleotide variation resulting in PTC is shown in red.

## Data Availability

The data presented in this study are available on request from the corresponding author.

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
