# Peer review of "Stop Codon Context-Specific Induction of Translational Readthrough"

_biomolecules, 2021, doi:10.3390/biom11071006_

Round 1
Reviewer 1 Report
The article by Schilff et al. reports the evaluation of the efficacy of readthrough promoters as a function of the genetic context surrounding the nonsense mutation. In general, the reported results lack the novelty required for these types of studies, particularly with reference to the effect of the genetic context on the readthrough efficacy of aminoglycosides. Several articles (see for instance Dabrowskj et al 2015) have mentioned such effect and in this paper, there is no discussion about the mechanism of action supporting the rationale for the observed effects. The manuscript cites as many as 49 references but only 5 of them are very recent (4 from 2020 and 1 from 2019) with a notable gap about literature 2015-2018, a period in which several studies have focused on the mechanism of action of TRIDs.
As for the methodology, the choice of cytofluorimetry for quantitative study in this particular field could be risky, especially when the percentage of readthrough is well below 10%. Indeed, reported data show very wide error bars that make questionable the significance of observed differences between mutations subjected to readthrough. Actual cytofluorimetric images/panel should accompany all data reported, at least in supplementary info.
As for the introduction, the paragraphs appear to be inhomogeneous without a flowing connection (see, for instance, but not only, lines 71-72 from SCC to peroxisomes).
As for the results, at line 154 the authors mention “several studies show that…”, but there is no citation at the end of the quote. The sentences at lines 165-168 should be rewritten for clarity. Moreover, figure 1F shows a very wide error bar for the mutation c. 1258 C>T making any conclusion about this determination very questionable. A similar situation is observable in figure 2-G.
At lines 176-179 the fact that C (+4) provokes a higher readthrough than G (+4) is known (see Welch, 2007 and Dabrowskj 2015) [notwithstanding the wide uncertainty of the measurements].
In line 186 the authors claim a dose-dependent readthrough, however, the plateau is reached at low doses in almost all the cases thus there is no linear correlation between dose and readthrough (maybe in one plot only). Additionally, in figure 2 caption 8 concentration points are mentioned while there are not 8 points in the graphs.
Importantly, in figure 1 C-D data refer to gentamicin while figures 1 EFG report data with G418 (which is a different compound). The untreated panel should precede the treated panel and positive control is lacking to show how RFP and GFP fluorescence is revealed. Moreover, it is not clear why (or how) RFP (PE-A filter) is higher in untreated cells. Panels should be also visualized without “gate”. Moreover, cells have been treated with penicillin and streptomycin and it is not clear whether these antibiotics were washed out before treatment.
On these bases, in my opinion, the manuscript does not reach the quality required for publication in Biomolecules.
The discussion often repeats parts of the introduction and does not cite recent literature on TRIDs. Moreover, in line 297, “PTC mutations” is not a correct definition and the entire sentence is not properly phrased. Line 364, reports comments that do not add useful information to the discussion; moreover, they refer to G418 as TRID without mentioning alternative readthrough inducers (e.g. Ataluren, ELX02, and others).
Author Response
The article by Schilff et al. reports the evaluation of the efficacy of readthrough promoters as a function of the genetic context surrounding the nonsense mutation. In general, the reported results lack the novelty required for these types of studies, particularly with reference to the effect of the genetic context on the readthrough efficacy of aminoglycosides. Several articles (see for instance Dabrowskj et al 2015) have mentioned such effect and in this paper, there is no discussion about the mechanism of action supporting the rationale for the observed effects. The manuscript cites as many as 49 references but only 5 of them are very recent (4 from 2020 and 1 from 2019) with a notable gap about literature 2015-2018, a period in which several studies have focused on the mechanism of action of TRIDs.
Response: We thank the reviewer for critical reading, comments and suggestions on the manuscript - these helped us immensely in the revision and improvement of the paper. In this study, we wanted to focus our analysis especially on PEX5 PTCs and their readthrough induction potential. The main aim of our work is to achieve a quantitative assessment of the effect of the stop codon context on readthrough stimulation. Analysis of the underlying mechanism of action would definitely be of interest, but is not within the scope of our study and should be analysed in a follow-up study.
We are now citing the suggested review article by Dabrowski et al. 2015 and some more recent publications.
We know added new data derived from our previously designed computational model for TR prediction. In our previous publications we developed a TR prediction tool that uses an in-silico regression analysis of all transcripts of the human transcriptome and experimental data TR data to calculate the readthrough propensity (Schueren et al. 2014, Hofhuis et al. 2016). We now used this model to predict TR values for the five PEX5 PTCs and compare these results with our experimental data so that we can discuss the properties of the readthrough-prone SCC in more detail.
One of our conclusions is that the knowledge of the stop codon context is a prerequisite (but not a sufficient condition) for optimal readthrough induction. Of note, the results of this study can principally be applied to other genes as they are not specific for PEX5.
As for the methodology, the choice of cytofluorimetry for quantitative study in this particular field could be risky, especially when the percentage of readthrough is well below 10%. Indeed, reported data show very wide error bars that make questionable the significance of observed differences between mutations subjected to readthrough. Actual cytofluorimetric images/panel should accompany all data reported, at least in supplementary info.
Response: Cytometry-based dual reporter assays have many advantages over, e.g. enzyme-based assays. Cells can be measured directly and do not have to be lysed, measurement and readthrough calculation can be done on the single-cell level and TRIDs do not affect the measurement as it is often the case with enzyme-based methods. Therefore, we decided to use this assay that we established and validated already some years ago (Hofhuis et. al., 2017). In this publication, we also compared our assay to other readthrough assays. We now prepared all the flow cytometry panels of the 450 or so measurements and made them available to the reviewers. To the reader, all primary data is available upon request.
The (error) bars show minimal to maximal values measured. This is the reason why they are in some cases very large. This can be visually overcome if presenting errors bars as “standard error of mean”. One can “improve” the presentation by showing SEM, however, it has no effect on the statistical analysis.
As for the introduction, the paragraphs appear to be inhomogeneous without a flowing connection (see, for instance, but not only, lines 71-72 from SCC to peroxisomes).
Response: We thank the reviewer for this comment. We re-wrote parts of the introduction for better flow.
As for the results, at line 154 the authors mention “several studies show that…”, but there is no citation at the end of the quote. The sentences at lines 165-168 should be rewritten for clarity. Moreover, figure 1F shows a very wide error bar for the mutation c. 1258 C>T making any conclusion about this determination very questionable. A similar situation is observable in figure 2-G.
Response: We added the citations. As mentioned in the earlier comment, the error bars represent min to max values. This does not affect the statistical analysis or the conclusions.
At lines 176-179 the fact that C (+4) provokes a higher readthrough than G (+4) is known (see Welch, 2007 and Dabrowskj 2015) [notwithstanding the wide uncertainty of the measurements].
Response: We have now mentioned the point and included the suggested citation (lines 180-182). It is, however, worth mentioning that TR inducibility (TR induction factor) for c1279 C>T with G (+4) is comparable to c1258 C>T with C (+4). These results mean that G418 treatment potentially is as advantageous in c1279 C>T as it is for c1258 C>T (see the lines 188-199).
In line 186 the authors claim a dose-dependent readthrough, however, the plateau is reached at low doses in almost all the cases thus there is no linear correlation between dose and readthrough (maybe in one plot only). Additionally, in figure 2 caption 8 concentration points are mentioned while there are not 8 points in the graphs.
Response: The reviewer is correct that noticing that there is not a linear correlation between TRID concentrations and readthrough. Rather at high doses a plateau is reached (“maximum attainable readthrough”). In Figure 2, we quantify these plateaus. We were surprised, how different these plateaus are, depending on the SCC, again underlining the strong SCC-dependence of readthrough.
The reviewer is right that for the SCCs c.826C>T, c.1279C>T and c.1090C>T only five concentrations (0, 10, 30, 70, 170 µg/ml) where measured. Eight concentration points were only measured for c.397C>T and c.1258C>T. Here, additionally 5, 20 and 100 µg/ml where used, as these SCCs already showed relatively high TR induction for the low concentration of 10 µg/ml. We changed this now in the text.
Importantly, in figure 1 C-D data refer to gentamicin while figures 1 EFG report data with G418 (which is a different compound). The untreated panel should precede the treated panel and positive control is lacking to show how RFP and GFP fluorescence is revealed. Moreover, it is not clear why (or how) RFP (PE-A filter) is higher in untreated cells. Panels should be also visualized without “gate”. Moreover, cells have been treated with penicillin and streptomycin and it is not clear whether these antibiotics were washed out before treatment.
Response: We thank the reviewer for this comment. We changed Figure 1 accordingly. RFP fluorescence is not higher in the untreated cells, it is only shifted due to the additional GFP signal. We did not wash out penicillin/streptomycin before measurement. We have these antibiotics also in the untreated samples, and we can exclude that these antibiotics influence readthrough measurement.
On these bases, in my opinion, the manuscript does not reach the quality required for publication in Biomolecules.
The discussion often repeats parts of the introduction and does not cite recent literature on TRIDs. Moreover, in line 297, “PTC mutations” is not a correct definition and the entire sentence is not properly phrased. Line 364, reports comments that do not add useful information to the discussion; moreover, they refer to G418 as TRID without mentioning alternative readthrough inducers (e.g. Ataluren, ELX02, and others).
Response: We changed the discussion accordingly and cite additional literature on TRIDs. We now refer also to other readthrough inducers.
Reviewer 2 Report
The authors present on five reporters, each is bearing a PEX5 PTC with 10 surrounding nucleotides; that the stop context has a strong influence on the levels of readthrough stimulation and impacts the choice of the most effective drug and its concentration. These results are in accord with most studies dealing with readthrough, stop codon context, PTC and TRIDs (over 20 years of research). That means that the conclusions in general are not striking, however for the specific Peroximal Biogenesis Disorders (PBD) research it will be highly valuable, because it directs this subgroup towards option of translational readthrough therapy. The authors also correctly outline what kind of future research needs to be done in order to estimate whether the readthrough stimulation will be a therapeutic option for PBD.
Minor comments
p.4, l.176. … typo – UAGC change to UGAC
p.9, l.330 … suggestion to change “PBD” to Peroximal Biogenesis Disorders
Author Response
Response: We thank the reviewer for the positive feedback. We made the changes accordingly.
Reviewer 3 Report
This is a nice study assessing the interplay between stop codon context and selected TRIDs. It is very nice that the authors identified 5 different disease-causing mutations in the same mRNA. The experiments have been well designed and well executed and the manuscript is well written. I am in favour of acceptance with only one minor correction (below). However, I have one suggestion that would help the reader. It would be nice to have the relevant figures labelled with the stop codon and 3’ nt instead of the actual mutation i.e. UGA_G instead of c.826C>T.
Lines 158-161 – There is a combination of ‘T’ and ‘U’ in the same sequence.
Author Response
Response: We thank the review for the positive feedback. We corrected the spelling accordingly and show now the SCC from nucleotide -3 to +7 in the figures.
Round 2
Reviewer 1 Report
The authors have thoroughly responded to comments and improved manuscript quality.